# Exclusive breastfeeding can attenuate body-mass-index increase among genetically susceptible children: A longitudinal study from the ALSPAC cohort

Yanyan Wu[1,2], Stephen Lye[2], Cindy-Lee Dennis[3,4], Laurent Briollais[2,5]*

**1** Office of Public Health Studies, Myron B. Thompson School of Social Work, University of Hawai'i at Mānoa, Honolulu, Hawai'i, United States of America, **2** Lunenfeld-Tanenbaum Research Institute, Mount Sinai Hospital, Toronto, Ontario, Canada, **3** Lawrence S Bloomberg Faculty of Nursing, University of Toronto, Toronto, Ontario, Canada, **4** Li Ka Shing Knowledge Institute, St. Michael's Hospital, Toronto, Ontario, Canada, **5** Dalla Lana School of Public Health, University of Toronto, Toronto, Ontario, Canada

* laurent@lunenfeld.ca

**Data Availability Statement:** Data from the ALSPAC cohort is available to researchers according to processes outlined at http://www.bristol.ac.uk/alspac/researchers/access/ and is

## Abstract

Recent discoveries from large-scale genome-wide association studies (GWASs) explain a larger proportion of the genetic variability to BMI and obesity. The genetic risk associated with BMI and obesity can be assessed by an obesity-specific genetic risk score (GRS) constructed from genome-wide significant genetic variants. The aim of our study is to examine whether the duration and exclusivity of breastfeeding can attenuate BMI increase during childhood and adolescence due to genetic risks. A total sample of 5,266 children (2,690 boys and 2,576 girls) from the Avon Longitudinal Study of Parents and Children (ALSPAC) was used for the analysis. We evaluated the role of breastfeeding (exclusivity and duration) in modulating BMI increase attributed to the GRS from birth to 18 years of age. The GRS was composed of 69 variants associated with adult BMI and 25 non-overlapping SNPs associated with pediatric BMI. In the high genetic susceptible group (upper GRS quartile), exclusive breastfeeding (EBF) to 5 months reduces BMI by 1.14 kg/m$^2$ (95% CI, 0.37 to 1.91, $p$ = 0.0037) in 18-year-old boys, which compensates a 3.9-decile GRS increase. In 18-year-old girls, EBF to 5 months decreases BMI by 1.53 kg/m$^2$ (95% CI, 0.76 to 2.29, $p$<0.0001), which compensates a 7.0-decile GRS increase. EBF acts early in life by delaying the age at adiposity peak and at adiposity rebound. EBF to 3 months or non-exclusive breastfeeding was associated with a significantly diminished impact on reducing BMI growth during childhood. EBF influences early life growth and development and thus may play a critical role in preventing overweight and obesity among children at high-risk due to genetic factors.

## Author summary

Previous studies have shown that EBF is associated with lower BMI during childhood and adolescence. Moreover, a GRS based on 97 genetic variants has been derived from large

consistent with the data-sharing policies of PLOS. Individual level data have restricted access and are distributed upon approval of research proposal and payment of data access costs.

**Funding:** This work was supported by the ALVA foundation, Toronto, (http://www.alva.ca/) (SL, LB), a team grant from the Accelerator Grants in genomic Medicine program of the McLaughlin Centre at University of Toronto (http://www.mclaughlin.utoronto.ca/) (LB), and by the National Institute on Minority Health and Health Disparities, National Institutes of Health, Ola HAWAI'I (Health And Wellness Achieved by Impacting Inequalities), grant #2U54MD007601-31 (YW). The funders had no role in study design, data collection and analysis, decision to publish, or preparation of the manuscript.

**Competing interests:** The authors have declared that no competing interests exist.

GWASs and is predictive of BMI in adults and children. However, it remains unclear whether EBF can attenuate the increase in BMI attributed to the GRS in children. Our study was able to characterize the effect of the GRS in children from birth to 18 years of age. Our main results showed that EBF to 5 months has substantial effect in decreasing BMI among children at higher genetic risks. EBF to 3 months or non-exclusive breastfeeding had a significantly diminished effect on reducing BMI growth during childhood. Our study suggests that interventions aimed at reducing the risks of overweight and obesity across the lifespan should start in very early childhood to be impactful, which makes EBF a key candidate intervention. While EBF is beneficial to all children, targeting those carrying multiple BMI/obesity alleles should be a priority to reduce obesity and associated non-communicable diseases.

## Introduction

Previous research has clearly established a link between early environments (prenatal and postnatal), genetic and behavioral factors on the developmental origins of health and disease (DOHaD) [1]. Among environmental factors, breastfeeding has been advocated in the prevention of overweight/obesity among children. The WHO suggests breastfeeding is the "perfect food for the newborn" and recommends all infants be exclusively breastfed up to 6 months of age, with continued breastfeeding along with appropriate complementary foods up to two years of age or beyond [2]. Importantly, there is growing evidence that breastfeeding may reduce the risk of being overweight [3]. A large meta-analysis from WHO showed that the odds of being obese among children who never breastfed or breastfed for less than 6 months vs. those who breastfed for at least 6 months were 1.22 (95% CI, 1.16 to1.28) for non-exclusive breastfeeding and 1.25 (95% CI, 1.17 to1.36) for EBF [4]. Despite numerous observational studies demonstrating the benefits of breastfeeding on a healthy infant growth, the biological functions underlying this effect are still poorly understood. It also remains unclear whether the beneficial effect of breastfeeding extends to children with higher genetic risks. Our previous analysis of the ALSPAC child cohort suggested that a longer duration of EBF (i.e. at least 5 months) has significant preventive effect on BMI growth trajectories among children carrying a genetic variant in the *FTO* gene [5]. Recently, a large GWAS based on 339,224 adult Caucasians identified 97 genetic variants strongly associated with BMI and explaining about 2.7% of BMI variability, which can be used to construct a GRS predictive of adult and children obesity-related traits [6]. This 97-SNPs GRS has been found to be associated with BMI across all ages in adults, with stronger associations in women than in men. This sex difference could reflect a greater heritability of adult BMI in women than in men, as reported in twin studies, or that different sets of genes influence adult BMI in men and women [7–9]. In terms of effect size, a 10-allele increment in the weighted GRS increases BMI by 1.35 kg/m$^2$ in women and 0.91 kg/m$^2$ in men, at 45 years of age [10]. In children, a similarly defined GRS was found associated with BMI at adiposity peak and childhood BMI, where a one-allele increment in the GRS increases BMI around 6 years of age by 0.112 kg/m$^2$ [11]. This GRS explained about 1.5% of child BMI variability at 6 years of age. Our previous work has shown that the effect of the GRS on pediatric BMI starts in early childhood and continues through adulthood [12]. While knowledge on the genetic architecture of adult and pediatric BMI is accumulating thanks to large-scale GWAS results, the construction of obesity-specific GRSs is emerging as an important approach for the personal and clinical management of individuals at risk of adverse outcomes [13, 14]. It is therefore timely to consider the protective effect of EBF among children

with elevated risk of overweight/obesity, where this risk is assessed by an obesity-specific GRS, and thus to extend our previous results on the *FTO* genetic variant. Our goal in this paper is to assess the effect of this GRS from infancy to the end of adolescence as well as the modulating effect of EBF during this time period.

## Methods

### Ethics statement

Ethical approval for the study was obtained from the ALSPAC Law and Ethics committee and our Local Research Ethics Board. Informed consent for the use of data collected via questionnaires and clinics was obtained from participants following the recommendations of the ALSPAC Ethics and Law Committee at the time. Please note that the study website contains details of all the data that are available through a fully searchable data dictionary and variable search tool [15]. Patients or the public WERE NOT involved in the design, or conduct, or reporting, or dissemination plan of our research.

### Cohort information

Our discovery cohort is the Avon Longitudinal Study of Parents and Children (ALSPAC) [16, 17]. Pregnant women resident in Avon, UK with expected dates of delivery 1st April 1991 to 31st December 1992 were invited to take part in the study. The core ALSPAC sample consists of 14,541 pregnancies. An additional 542 eligible pregnancies not in the core sample, who were invited to participate at age 7 and for whom research data were available in November 2004, were also included in our study. Overall, these 15,083 pregnancies resulted in 15,224 known fetuses. For reasons of confidentiality data on the 13 triplet and quadruplet children were not available for analysis. After removing children without anthropometric measures (height/length or weight, n = 2,462), non-Caucasian children (n = 2,314), those without genotype data (n = 3,537) or without exclusive breastfeeding (n = 857) or socio-economic information (n = 775), a total of 2,690 boys and 2,576 girls (N = 5,266) was available for our analyses. These children have been followed for over two decades. The description of the cohort is given in Table 1.

### Exclusive breastfeeding

Information pertaining to early infant feeding was collected. Mothers recorded the age at which breastfeeding was stopped (in months), and the age at which supplementation with milk other than breast milk was introduced (in months). This information was determined from the mother's diary of early feeding milestones, as well as from an interview with the study nurse at the 6-month child follow-up and survey questions at the 15-month child follow-up. The duration of EBF was defined as the provision of breastmilk only from the time of from birth until the introduction of other milk (non-breast milk) or any solid. Different coding strategies for EBF were assessed using either categorical or continuous variables. The most significant effect of EBF was obtained under a continuous coding, which can be interpreted as a dose-response relationship between BMI and EBF.

### Genetic risk score

We used 69 SNPs associated with BMI at genome-wide significance in the Genetic Investigation of Anthropometric Traits (GIANT) consortium and that were recently included in a gene-obesogenic interaction study [18] as well as 25 independent non-overlapping SNPs that we previously studied in relation to pediatric BMI trajectories to create a GRS of 94 SNPs (S1 Table), which represents the genetic susceptibility to overweight and obesity [12]. The sex-

**Table 1. Summary statistics for individual level variables and BMI measurements by age in years.** Chi-square or two-sample t-test was carried out to examine differences between boys and girls for individual level variables.

| | Individual-level variables | | | BMI measurement by age | | | | |
|---|---|---|---|---|---|---|---|---|
| | **Boys** | **Girls** | | **Age (year)** | **Boys** | | **Boys** | |
| | **N = 2690** | **N = 2576** | **p-value** | | **N** | **Mean(SD)** | **N** | **Mean(SD)** |
| *Categorical Variables* | **N(%)** | **N(%)** | | | | | | |
| Mother's Education | | | | Birth | 2013 | 13.9(1.8) | 1947 | 13.8(1.7) |
| CSE/none | 313 (11.6%) | 303 (11.8%) | 0.734 | 1 | 716 | 18.0(1.4) | 665 | 17.7(1.4) |
| Vocational | 227 (8.4%) | 193 (7.5%) | | 2 | 638 | 17.1(1.3) | 591 | 16.8(1.4) |
| O Level | 955 (35.5%) | 904 (35.1%) | | 3 | 641 | 16.6(1.3) | 601 | 16.5(1.5) |
| A Level | 720 (26.8%) | 711 (27.6%) | | 4 | 676 | 16.4(1.3) | 634 | 16.3(1.6) |
| Degree | 475 (17.7%) | 465 (18.1%) | | 5 | 719 | 16.0(1.6) | 682 | 16.0(1.7) |
| Mother's pregnancy smoking status | | | | 6 | 1898 | 15.7(1.6) | 1796 | 15.6(1.8) |
| Never | 1493 (55.5%) | 1450 (56.3%) | 0.388 | 7 | 1361 | 16.1(1.9) | 1286 | 16.3(2.1) |
| No during pregnancy | 691 (25.7%) | 679 (26.4%) | | 8 | 1819 | 16.4(2.0) | 1790 | 16.7(2.2) |
| Yes during pregnancy | 506 (18.8%) | 447 (17.4%) | | 9 | 1074 | 17.0(2.3) | 1044 | 17.4(2.5) |
| Mean family income per week | | | | 10 | 2917 | 17.6(2.8) | 3006 | 17.9(2.9) |
| < £100 | 45 (1.7%) | 48 (1.9%) | 0.634 | 11 | 1338 | 18.1(3.0) | 1281 | 18.4(3.1) |
| < £200 | 329 (12.2%) | 309 (12.0%) | | 12 | 2044 | 18.9(3.3) | 2119 | 19.2(3.3) |
| < £300 | 479 (17.8%) | 473 (18.4%) | | 13 | 1675 | 19.4(3.4) | 1750 | 20.0(3.4) |
| < £400 | 1038 (38.6%) | 946 (36.7%) | | 14 | 1745 | 19.9(3.3) | 1826 | 20.6(3.4) |
| ≥ £400 | 799 (29.7%) | 800 (31.1%) | | 15 | 1058 | 20.9(3.3) | 1116 | 21.6(3.5) |
| | | | | 16 | 449 | 21.1(3.3) | 512 | 21.8(3.5) |
| *Continuous Variables* | **Mean (SD)** | **Mean (SD)** | | 17 | 226 | 22.2(3.5) | 273 | 22.2(3.6) |
| Mother's pre-pregnancy BMI | 23.0 (3.8) | 22.9 (3.8) | 0.353 | 18 | 989 | 22.5(3.9) | 1236 | 22.9(4.1) |
| Duration of EBF (month) [a] | 1.6 (1.56) | 1.7 (1.59) | 0.004 | 19–20 | 67 | 22.5(3.3) | 79 | 23.2(4.0) |
| Duration of BF (month) [b] | 4.6 (4.71) | 4.9 (4.66) | 0.090 | | | | | |
| Gestational age (weeks) | 39.5 (1.8) | 39.6 (1.7) | 0.006 | | | | | |
| GRS (Range 0–10) [c] | 5.0 (1.3) | 5.0 (1.3) | 0.520 | | | | | |
| GRS (number of risk alleles) [d] | 95.3 (6.7) | 92.7 (7.1) | <0.0001 | | | | | |
| (min, max) | (73, 119) | (68, 120) | | | | | | |

[a] Duration of EBF (exclusive breastfeeding in months).

[b] Duration of BF (non-exclusive breastfeeding in months).

[c] GRS (genetic risk score, deciles) were derived for boys and girls separately.

[d] GRS in raw scales (number of risk alleles). 1-decile increase in the GRS corresponds to a 4.6-allele effect in boys and 5.2 allele-effect in girls.

specific GRSs were created using the imputed dosages for the 94 SNPs where each SNP was recoded to represent the number of BMI-increasing alleles and was weighted using the sex-specific weights derived from the GIANT consortium and UK BioBbank meta-analysis [19] and available through the portal: https://portals.broadinstitute.org/collaboration/giant/index.php/GIANT_consortium_data_files. GRS scores were then created for boys and girls separately by scaling the sum of the weighted SNP effects ($\Sigma \beta_i \times SNP_i$, $i = 1, \ldots, 94$) to a range of 0 to 10. With this transformation, a 1-unit (i.e. 1-decile) increase in the GRS corresponded to a 4.6-allele effect in boys and 5.2 allele-effect in girls.

## Assessment of BMI and control variables

Birth length (crown-heel) was measured by ALSPAC staff who visited newborns soon after birth (median 1 day, range 1–14 days), using a Harpenden Neonatometer (Holtain Ltd). Birth

weight was extracted from medical records. From birth to five years, length and weight measurements were extracted from health visitor records, which form part of standard childcare in the UK. In this cohort we had up to four measurements taken on average at six weeks and at 10, 21, and 48 months of age. For a random 10% of the cohort, we also have length/height measurements from eight research clinic visits, held between the ages of four months and five years of age. From age seven years upwards, all children were invited to annual clinics. In addition, parent-reported child height and weight were also available from the questionnaires. BMI was derived from height and weight measurements (mean 9 measurements per individual) and calculated as the weight (in kg) divided by the square of height (in cm). The following confounding variables consistently associated with breastfeeding were controlled in the analysis: gestational age (in months), maternal preconception BMI, education and smoking status, and family income. The gestational age was calculated based on a variety of records including last menstrual period, pediatric assessment, obstetric assessment and ultrasound assessment. Self-reported maternal preconception BMI was collected from the "About Yourself" questionnaire at 12 weeks of gestation. Maternal education status was obtained from the "Your Pregnancy" questionnaire administered at 32-weeks of gestation and coded as: Certificate of Secondary Education (CSE)/none; vocational; O level; A level and Degree. Maternal smoking status was collected from the "Having a Baby" questionnaire at 18-week gestation and was coded as: Never; Yes during pregnancy; Not during pregnancy. Family income was collected at the 33, 47, 85, 97, 134 months and 18 years follow-up visits and the mean weekly income was categorized into one of five levels: less than £100, £100–£199, £200–£299, £300–£399, and £400 per week or more. Gestational age and maternal preconception BMI were centered at the means and analyzed as continuous variables. The levels with the largest proportions for categorical variables were used as the reference groups in the analysis.

## Statistical methods

Summary statistics were used to describe the sample characteristics for boys and girls. A mixed-effects model approach with cubic splines of age (S1 Text) was used to fit the longitudinal BMI data from the ALSPAC cohort from birth to 20 years of age in boys and girls separately [20]. We examined three-way interactions between cubic splines of age, EBF and GRS. Both backward elimination and stepwise variable selections were used to select the best model and optimal spline knots. We calculated the predicted BMI trajectories (i.e., the population average) up to age 18 years with GRS scores evaluated at the three quartiles 2.5, 5.0 and 7.5 for zero and five months of EBF, respectively, to characterize the effect of GRS and EBF on BMI trajectories. Hypothesis testing of GRS and EBF effect at specific ages was performed by using the generalized linear hypothesis (GLM) approach (S1 Text) [21]. We also estimated the timing of adiposity peak (AP) and adiposity rebound (AR). The bootstrap method with 2,000 iterations was used to test the effect of GRS and EBF on AR and AP [22]. Additionally, we replaced the EBF variable with non-exclusive BF to examine if EBF had stronger effect than any BF.

Statistical analyses were performed using the statistical software *R* 3.5.1. Statistical packages in R include "nlme", "effects", "spida2" and "ggplot2". All hypothesis tests were 2-sided and the priori level of significance was set at 5%.

## Missing data

Children with missing longitudinal BMI observations over time were included in our analyses as long as they had at least one BMI observation available between birth and 20 years. The

estimation from mixed-effects models remains valid in that situation assuming the longitudinal observations are missing at random [20].

## Results

### Effect of the GRS on pediatric BMI growth trajectories

The GRS is associated with higher BMI with an increasing effect with age (Fig 1 and S2 Table). A quartile (2.5 units) increment in the GRS increases BMI by 0.61 kg/m$^2$ (95% CI, 0.47 to 0.75, $p<0.0001$) at age 7 years and 1.98 kg/m$^2$ (95% CI, 1.65 to 2.32, $p<0.0001$) at age 18 years among boys. The corresponding effects in girls are 0.39 kg/m$^2$ (95% CI, 0.24 to 0.55, $p<0.0001$) and 0.75 kg/m$^2$ (95% CI, 0.40 to 1.09, $p<0.0001$). These effects become significant from 5 years of age.

### Effect of GRS on the timing of adiposity peak (AP) and adiposity rebound (AR)

The GRS had no significant effect on the age at AP but was negatively associated with the age at AR among boys and girls, where a higher level of GRS corresponds to earlier age at AR (S3 Table). For instance, a GRS score of 5.0 vs. 2.5 (median vs. 1$^{st}$ quartile) advances the age at AR by 0.36 years (95% CI, 0.37 to 0.46, $p<0.0001$) and a GRS score of 7.5 vs. 2.5 (inter-quartile difference) by 0.65 years (95% CI, 0.49 to 0.80, $p<0.0001$) in boys. These effects in girls are 0.31 year (95% CI, 0.21 to 0.41, $p<0.0001$) and 0.57 year (95% CI, 0.39 to 0.741, $p<0.0001$), respectively.

### Effect of EBF on child longitudinal BMI by GRS levels

Our results indicate a significant 3-way interaction between age, GRS and EBF (or BF) in boys and girls (S4 Table). EBF has a stronger protective effect as the children become older and the

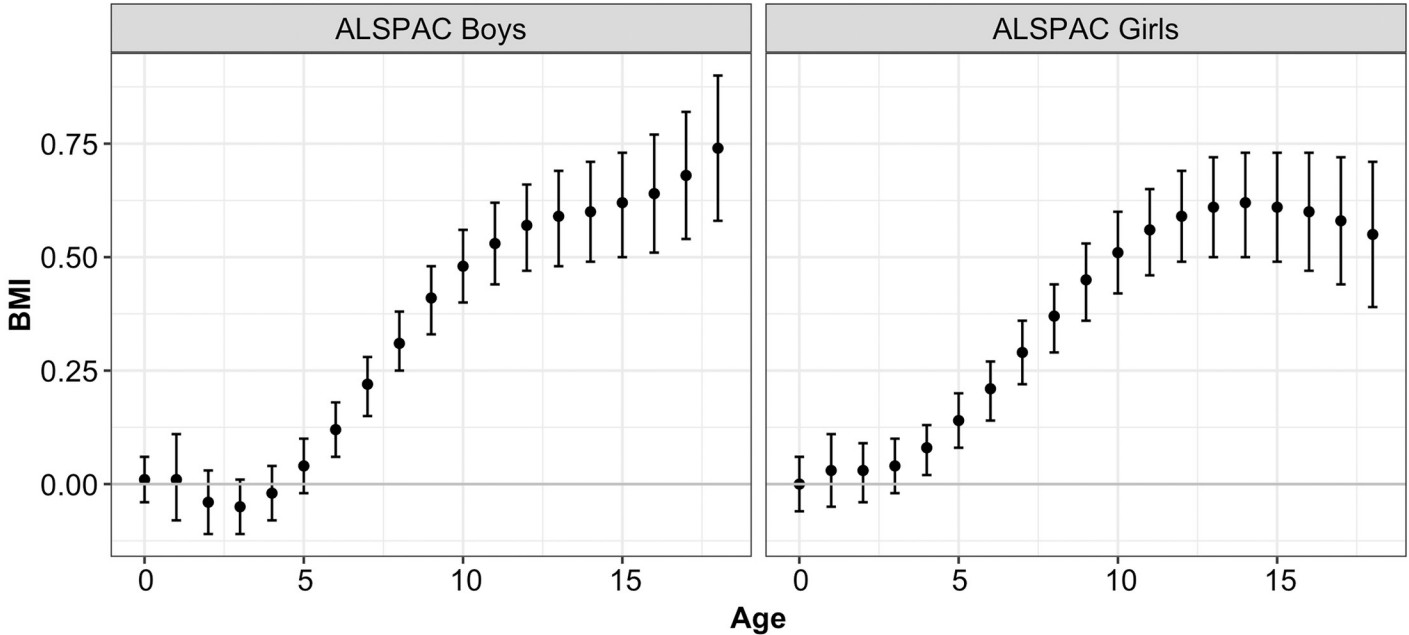

**Fig 1. Marginal effect of 2.5 units increase in GRS on pediatric BMI from birth to 18 years of age for boys and girls.**

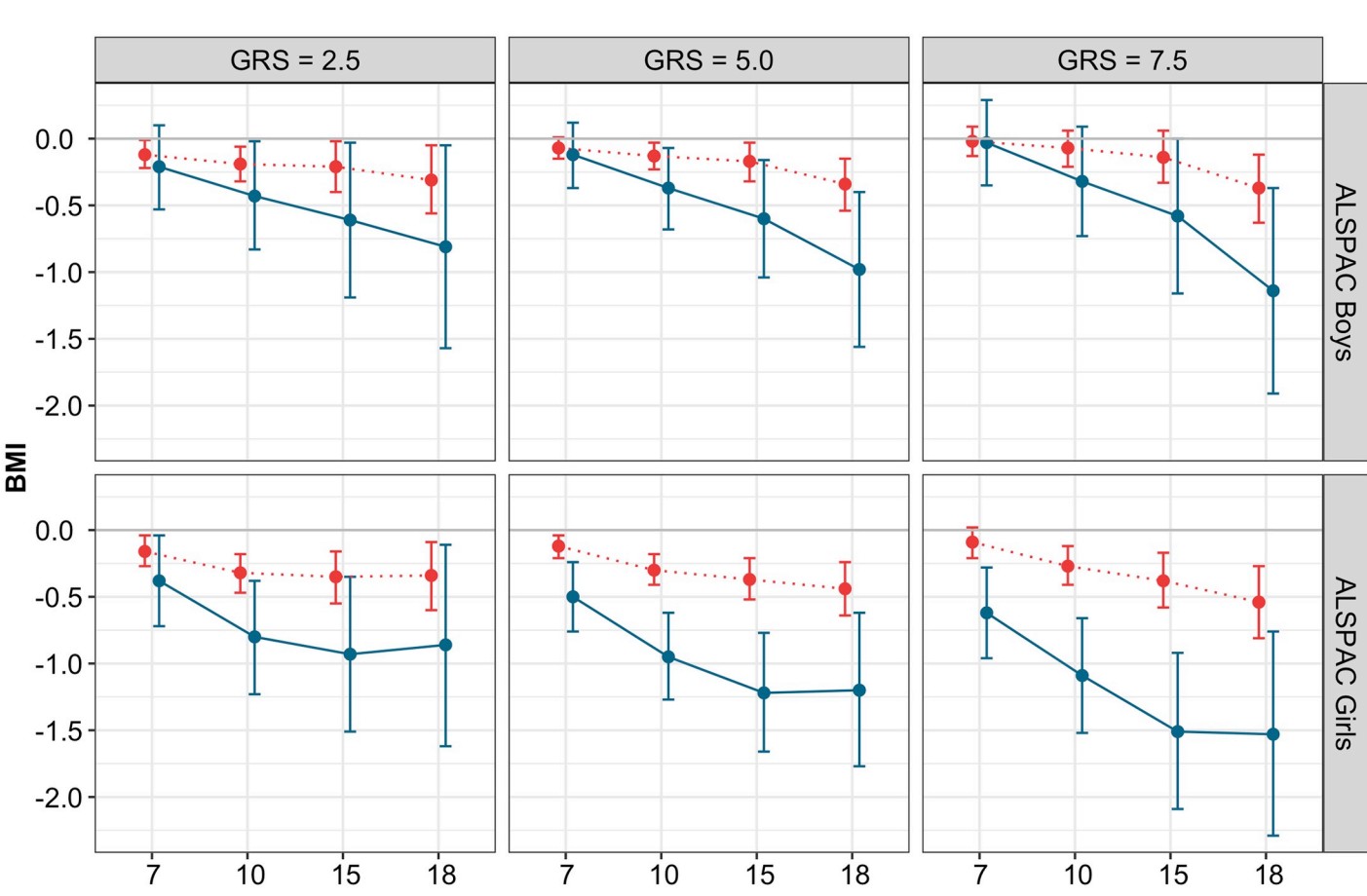

**Fig 2. Effect of 5 months of exclusive breastfeeding (EBF) and non-exclusive breastfeeding (BF) on BMI measurements at 7, 10 15 and 18 years of age for GRS scores evaluated at 2.5, 5.0 and 7.5.**

effect is greater with increasing GRS (Fig 2, S5 Table). In boys, at the first quartile of GRS (GRS = 2.5), five-month EBF decreases BMI by 0.21 kg/m$^2$ ($p = 0.19$) at 7 years and 0.81 kg/m$^2$ (95% CI, 0.05 to 1.57, $p = 0.0362$) at 18 years. At the median GRS level (GRS = 5.0), this BMI decrease is 0.12 kg/m$^2$ ($p = 0.32$) and 0.98 kg/m$^2$ (95% CI, 0.40 to 1.56, $p = 0.001$). At the third quartile of GRS (GRS = 7.5), this decrease reaches 0.03 kg/m$^2$ ($p = 0.85$) and 1.14 kg/m$^2$ (95% CI, 0.37 to 1.91, $p = 0.0037$), respectively. In girls, five-month EBF decreases BMI by 0.38 kg/m2 (95% CI, 0.04 to 0.72, $p = 0.0272$) at 7 years and 0.86 kg/m$^2$ (95% CI, 0.11 to 1.62, $p = 0.0252$) at 18 years at the first GRS quartile. This decrease reaches 0.50 kg/m$^2$ (95% CI, 0.24 to 0.76, $p = 0.0002$) and 1.20 kg/m$^2$ (95% CI, 0.62 to 1.77, $p<0.0001$) at the median GRS level, 0.62 kg/m$^2$ (95% CI, 0.28 to 0.96, $p = 0.0003$) and 1.53 kg/m$^2$ (95% CI, 0.76 to 2.29, $p<0.0001$) at the third quartile GRS level for age 7 and 18 respectively.

## Effect of 5 months EBF on timing of AP and AR by GRS levels

EBF to 5 months delays the age of AP significantly in boys with the average/high levels of GRS: 0.21 year (95% CI, 0.05 to 0.36, $p = 0.0076$) and 0.25 year (95% CI, 0.05 to 0.42, $p = 0.0136$) for GRS level of 5.0 and 7.5, respectively (S6 Table). A shorter delay in the age at AP was observed

in girls, i.e. 0.14 year (95% CI, 0.04 to 0.24, $p$ = 0.0063) and 0.24 year (95% CI, 0.09 to 0.38, $p$ = 0.0011), respectively. A duration of 5 months of EBF delays also the age at AR significantly in girls all levels of GRS, i.e. 0.64 years (95% CI, 0.15 to 1.16, $p$ = 0.0114), 0.53 years (95% CI, 0.20 to 0.86, $p$ = 0.0015), and 0.44 (95% CI, 0.05 to 0.85, $p$ = 0.0278) for GRS of 2.5, 5.0 and 7.5, respectively. It delays also the age at AR in boys but to a lesser extent and not significantly.

### Effect of non-exclusive BF on pediatric BMI growth trajectories

The effect of non-exclusive BF had less impact on BMI growth trajectories at different ages compared to the effect of EBF (Fig 3, S5 Table). For instance, at 18 years, the reduction of BMI associated with 5 months of non-exclusive BF varied in boys from 0.31 (95% CI, 0.05 to 0.56, $p$ = 0.0172) to 0.37 (95% CI, 0.12 to 0.63, $p$ = 0.0042) between the first and third GRS quartiles, and from 0.34 (95% CI, 0.09 to 0.60, $p$ = 0.0075) to 0.54 (95% CI, 0.27 to 0.81, $p$<0.0001) in girls.

### Dose-response relationship of EBF duration on pediatric BMI

As expected, a duration of EBF for 3 months had significantly less impact in decreasing BMI than 5 months of EBF (S7 Table, S1 Fig), and BF 3 month and 5 months had less effect compared to EXBF. At 18 years, the range of variation was -0.49 kg/m$^2$ to -0.68 kg/m$^2$ in boys and from -0.52 kg/m$^2$ to -0.92 kg/m$^2$ in girls, respectively, across the GRS categories (Fig 4). A duration of 3-months EBF also had a decreased influence on delaying the age of AP and AR compared to a 5-months duration (S6 Table). This is an important result since rapid weight gain during infancy is known to predispose to later onset of overweight and obesity during adulthood.

## Discussion

Our study demonstrates the role of the duration and exclusivity of breastfeeding in reducing BMI increases during childhood and adolescence resulting from adverse genetic effects. In the high genetic susceptible group (upper GRS quartile), EBF to 5 months reduces BMI by 1.14 kg/m$^2$ (95% CI, 0.37 to 1.91, $p$ = 0.0037) in 18-year-old boys, which compensates a 3.9-decile GRS increase. In 18-year-old girls, EBF to 5 months decreases BMI by 1.53 kg/m$^2$ (95% CI, 0.76 to 2.29, $p$<0.0001), which compensates a 7.0-decile GRS increase. EBF acts early in life by delaying the age at AP and at AR. Importantly, EBF to 3 months and non-exclusive breastfeeding to 5 months had a significantly less effect on BMI clearly demonstrating a strong dosage effect of continued EBF. These results reiterate the importance of EBF to 6 months as recommended by WHO.

The role of the obesity-specific GRS has been recently studied in children and adolescents [10–13, 23–28] and recent evidence suggests a continuum of risks starting from early childhood [12] and rising up to the mid 40s [10]. Our study confirms this trend and helps better characterize the GRS effect during childhood, showing a clear increasing trend from early infancy to late adolescence/early adulthood in boys and girls. Our results also shed light into the critical role of EBF in early development by showing how it delays the age at AP and AR and brings new insights by emphasizing that its effect in the high-susceptible genetic group is more substantial right after the timing of AR. During this developmental period, BMI is a strong predictor of later overweight/obesity development [29–31].

Recent efforts have demonstrated the clinical utility of the GRS in predicting overweight and obesity risks [13, 14]. A recent paper using an extended version of the GRS based on 2.1 million genetic variants, stressed the greatly increased risk of severe obesity among individuals in the top decile of the GRS. For instance, 15.6% of individuals in the top decile of GRS went

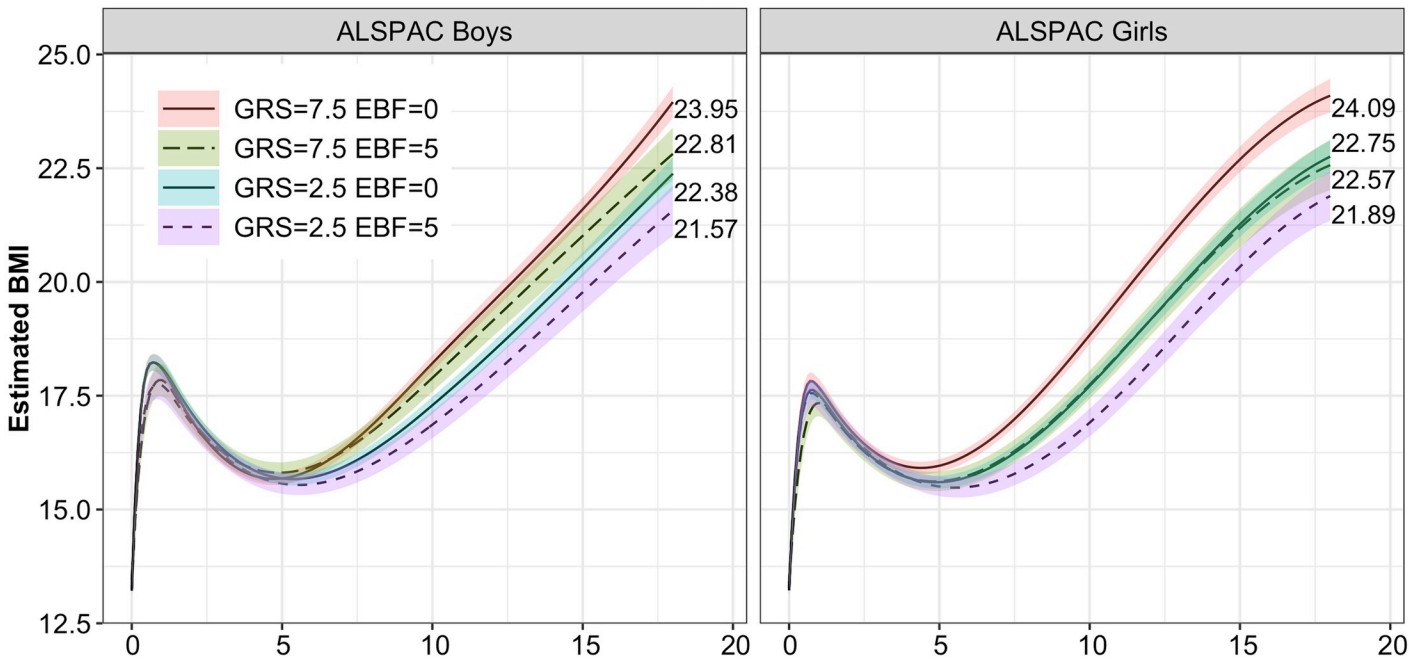

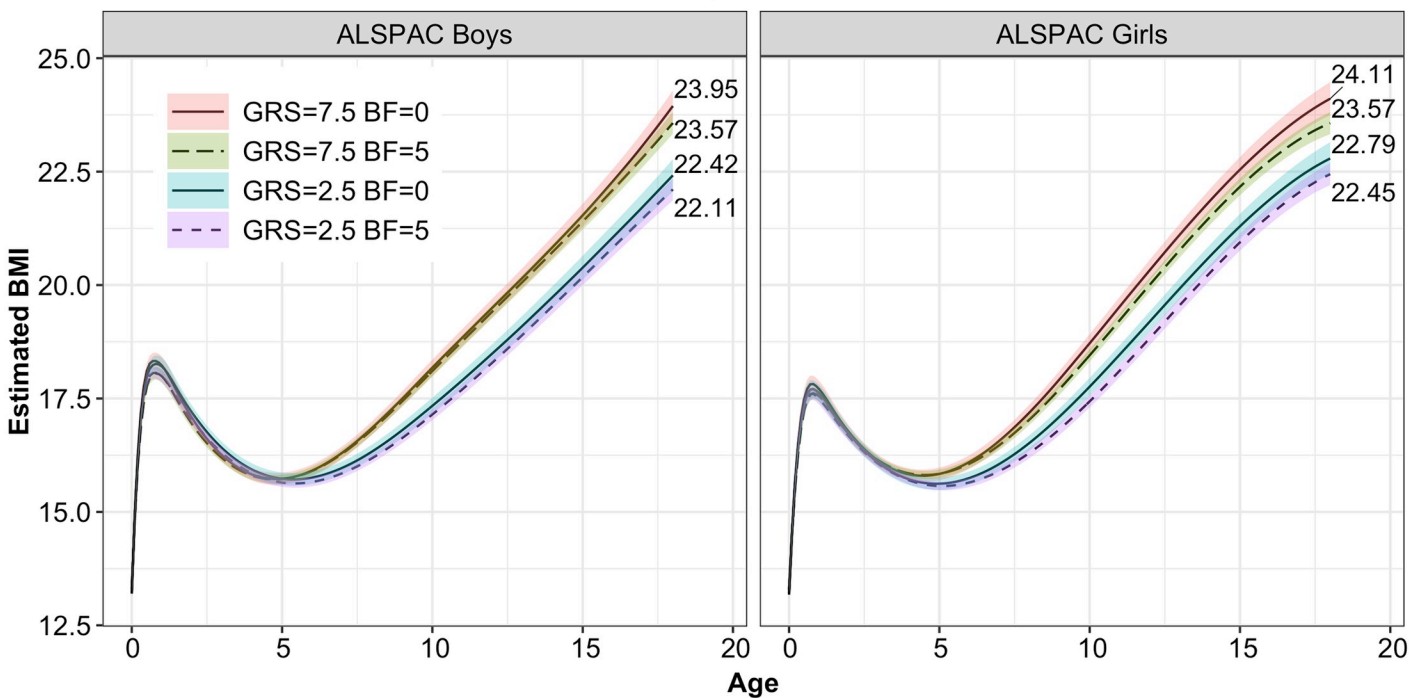

**Fig 3. Predicted BMI growth trajectories for ALSPAC boys and girls from birth to age 18 years for GRS = 2.5 and 7.5, and (a) EBF = 0 or 5 months, and (b) BF = 0 or 5 months.**

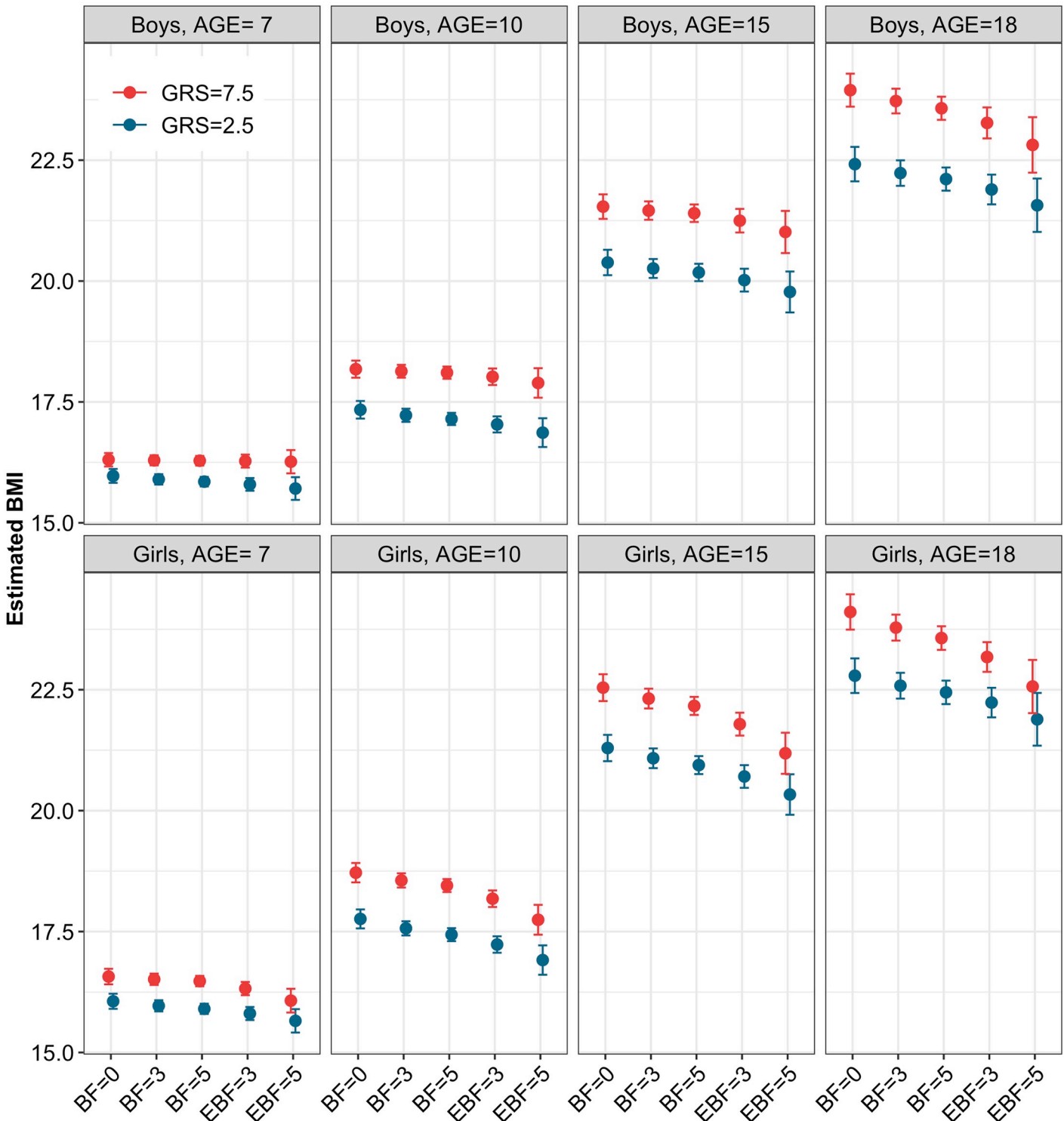

**Fig 4. Effect sizes of GRS on BMI and attenuation effects of 3 and 5 months of non-exclusive breastfeeding (BF) and exclusive breastfeeding (EBF) among ALSPAC boys and girls at 7, 10, 15 and 18 years.** In each sub-figure are represented the GRS effect on BMI (first 2 bars from the left) and the attenuation effect of 3-months BF (second 2 bars from the left), 5-months BF (third 2 bars from the left), 3-months EBF (second 2 bars from the right) and 5-months EBF (last 2 bars from the right).

on to develop severe obesity compared with 5.6% of those in deciles 2–9 and 1.3% in the lowest decile [13]. This top decile of the population had also a 4.2, 6.6 and 14.4-fold increased risk of a high BMI of 40, 50 and 60 compared to the rest of the population and also had increased risks of cardiometabolic diseases and overall mortality. Targeting this 10% decile population might therefore offer a cost-efficient strategy to reduce obesity-related morbidity, although this would need to be thoroughly evaluated. These authors also stressed the importance of early intervention, acknowledging that "given that the weight trajectories of individuals in different GRS deciles start to diverge in early childhood, such interventions may have maximal effect when employed early in life." As noted also in Torkamani et. al. [14], a targeted intervention might help "clarifying a high-risk individual's perception of their susceptibility to disease and quantifying the benefits of healthy behaviors could be an effective tool to induce and maintain behavioral changes".

Our study suffers from a number of limitations. Due to the relatively short duration of EBF in ALSPAC, we were not able to assess the effect of more than 5 months of EBF on pediatric BMI trajectories. Our GRS definition was based on 94 SNPs including 69 SNPs from Locke et. al., 2015 [6]. As large meta-GWASs on BMI and obesity-related traits are fast developing, some extended definitions of GRSs are emerging such as Yengo et.al. 2018 [19] and Khera et.al, 2019 [13]. However, the correlation between BMI and GRS in the Health and Retirement Study participants derived from Locke at.al., 2015 is near identical to the correlation between the BMI and GRS derived from Yengo et.al., 2018 (r = 0.22) [32]. We are planning to generalize our study to these new GRS definitions in the near future. Also, our GRS definition is mainly based on SNPs found associated with BMI in adults and could be extended to include genetic variants more specific to children, taking advantage on recent GWAS discoveries [33–35].

Clinically, from a public health perspective, the promotion of EBF could play a pivotal role in the programming of healthy life trajectories since breast milk is the first postnatal nutritional environment of all mammals and is now widely recognized as essential for optimal infant growth and development [36]. There is now widespread acceptance that the health benefits of breastfeeding continue well into the early childhood and beyond. The benefits for women have also been highlighted [37]. The 2016 Lancet Breastfeeding Series quantified the impact of these health and development benefits on healthcare costs and economic growth reporting that increases in breastfeeding rates could save US$400 million in health care costs in the US, UK, Brazil and China alone and inject US$300 billion into economies from more a productive workforce [38]. Despite these enormous benefits, only 40%, or two out of every five, infants globally are exclusively breastfed to 6 months postpartum as recommended. Successful breastfeeding programs directed at women are thus needed to achieve a longer duration of exclusive breastfeeding which, according to our findings, should be an important part of a comprehensive overweight or obesity prevention program to promote healthy growth trajectories during infancy that continue later in life.

While the benefits of breastfeeding on a healthy infant growth are well demonstrated, the biological functions underlying this effect are still poorly understood. The protective effect of breastfeeding could stem from its micronutrients and bioactive composition. Another hypothesis suggests the lower protein content of human milk compared with formula milk as the source of this protective effect [39]. Understanding the biological mechanisms underlying the beneficial effect of breastfeeding on healthy growth warrants further investigations.

## Supporting information

**S1 Text. Supplementary methods.**
(DOCX)

**S1 Table. The list of 94 SNPs used for GRS calculation.** GRS scores were created for boys and girls separately by scaling the sum of the weighted SNP effects. The weights were beta coefficients obtained from stratified meta-analysis of genome-wide association studies of BMI for men and women in $\sim 700000$ individuals of European ancestry (GIANT consortium and the UK Biobank).
(DOCX)

**S2 Table. Marginal effect of 2.5 units increase in GRS on pediatric BMI from birth to 18 years of age.**
(DOCX)

**S3 Table. Marginal effect of 2.5 units increase of GRS score on the age at adiposity peak (AP) and the age at adiposity rebound (AR).** The 95% confidence intervals (CIs) are computed with the bootstrap method3 with 2,000 iterations.
(DOCX)

**S4 Table. Three-way interaction of cubic splines of age, GRS and breastfeeding (EBF: Exclusive breastfeeding, BF: Non-exclusive breastfeeding) for boys and girls.** The optimal knots were ($\mathcal{K}_1 = 0.7$, $\mathcal{K}_2 = 1.5$, $\mathcal{K}_3 = 10$) for boys and ($\mathcal{K}_1 = 0.9$, $\mathcal{K}_2 = 1.5$, $\mathcal{K}_3 = 10$) for girls. The knots were defined in supplementary file section A.2.
(DOCX)

**S5 Table. Effect of 5 months of exclusive breastfeeding (EBF) and non-exclusive breastfeeding (BF) on BMI at 7, 10 15 and 18 years of age for GRS scores evaluated at 2.5, 5.0 and 7.5.**
(DOCX)

**S6 Table. Effect of 3 months and 5 months of exclusive breastfeeding (EBF) on the age at adiposity Peak (AP) and the age at adiposity rebound (AR) for GRS scores evaluated at 2.5, 5.0 and 7.5 and overall EBF effect regardless of GRS.** The 95% confidence intervals (CIs) are computed with the bootstrap method with 2,000 iterations.
(DOCX)

**S7 Table. Effect of 3 months of exclusive breastfeeding (EBF) and non-exclusive breastfeeding (BF) on BMI measurements at 7, 10 15 and 18 years of age for GRS scores evaluated at 2.5, 5.0 and 7.5.**
(DOCX)

**S1 Fig. Predicted BMI growth trajectories for ALSPAC Boys and Girls from birth to age 18 for GRS = 2.5, 5.0, or 7.5, and (a) EBF = 0 or 3 months, and (b) BF = 0 or 3 months.**
(DOCX)

## Acknowledgments

We are extremely grateful to all the families who took part in this study, the midwives for their help in recruiting them, and the whole ALSPAC team, which includes interviewers, computer and laboratory technicians, clerical workers, research scientists, volunteers, managers, receptionists and nurses. The UK Medical Research Council and the Welcome Trust (Grant ref: 102215/2/13/2) and the University of Bristol provide core support for ALSPAC.

## Author Contributions

**Formal analysis:** Yanyan Wu.

**Funding acquisition:** Stephen Lye, Laurent Briollais.

**Investigation:** Yanyan Wu, Stephen Lye, Cindy-Lee Dennis, Laurent Briollais.

**Methodology:** Yanyan Wu, Laurent Briollais.

**Supervision:** Laurent Briollais.

**Writing – original draft:** Yanyan Wu, Laurent Briollais.

**Writing – review & editing:** Yanyan Wu, Stephen Lye, Cindy-Lee Dennis, Laurent Briollais.

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
