## [Decision Letter · Decision Letter 0]

26 Jan 2020

Dear Dr Briollais,

Thank you very much for submitting your Research Article entitled 'Exclusive breastfeeding can attenuate body-mass-index increase among genetically susceptible children: a longitudinal study from the ALSPAC cohort' to PLOS Genetics. Your manuscript was fully evaluated at the editorial level and by independent peer reviewers. The reviewers appreciated the attention to an important problem, but raised some substantial concerns about the current manuscript. Based on the reviews, we will not be able to accept this version of the manuscript, but we would be willing to review again a much-revised version. We cannot, of course, promise publication at that time.

If you decide to revise the manuscript for further consideration at PLOS Genetics, please aim to resubmit within the next 60 days, unless it will take extra time to address the concerns of the reviewers, in which case we would appreciate an expected resubmission date by email to plosgenetics@plos.org.

[LINK]

We are sorry that we cannot be more positive about your manuscript at this stage. Please do not hesitate to contact us if you have any concerns or questions.

Yours sincerely,

Samuli Ripatti

Associate Editor

PLOS Genetics

Gregory Barsh

Editor-in-Chief

PLOS Genetics

There are multiple concerns raised by the Reviewers needs to be addressed and that prevent publication of the manuscript in its current form, in particular: 1) Please clarify what is the novelty of the current manuscript compared to the previous papers from ALSPAC and other data, 2) Please clarify the potential overlap of samples between the PRS weights and the test cohort and use other source for weights if they exist, and 3) present a formal statistical test for interaction as the basis for inference.

Reviewer's Responses to Questions

**Comments to the Authors:**

Reviewer #1: Thank you for the opportunity to review an interesting manuscript from effects of exclusive of breast feeding (EBF) and genetics on BMI trajectories. The Results section is clearly written and describes the age- and sex dependent complex relationships very well. However, there are some points which require clarification, especially the sample used to derive the SNP weights.

1) What sample was used to derive the SNP weights for the GRS? Was there overlap with ALSPAC?

2) Increased maternal BMI is a risk factor for emergency caesarean section and mode of delivery is associated for example with long-term changes in gut microbiota compared to vaginal delivery. As approximately half of the GRS is inherited from mother and thus affects also maternal BMI, could mode of delivery be a confounding factor?

3) In the discussion, you suggest that obesity interventions could target subjects in the upper tail of GRS. What GRS percentile you would target? Could any intervention be effective enough to justify screening costs considering that by targeting upper tail most of the obesity cases are missed (in absolute numbers due to Rose paradox)? Why do you think that targeted intervention would be more reasonable approach than non-targeted one?

4) Table 3: In boys, the effect of EBF on BMI seems to be larger when GRS is high. In girls, it seems to be the opposite (for example in 15 yo -1.55 if GRS=2.5 and -0.87 if GRS=7.5). Any suggestions why?

Reviewer #2: The authors have performed a study on the association of exclusive breastfeeding with BMI, adiposity peak, and adiposity rebound in 5,266 children from the ALSPAC cohort, and the interaction of exclusive breastfeeding with a genetic risk score for increased BMI. The manuscript reports that exclusive breastfeeding reduces BMI in adolescents and attenuates the impact of a genetic risk score for BMI. While the results are interesting, the novelty of the study seems rather limited. The authors have previously published results from a similar analysis of the interaction between exclusive breastfeeding and the FTO obesity risk locus on BMI trajectories in the ALSPAC cohort [29040503]. The link between breastfeeding and child BMI has been previously studied in ALSPAC, by Brion et al. [PMID 21349903], although in somewhat less detail than here.

My other main comments are the following:

1. The authors have used a weighted GRS, but the way the variants have been weighted when constructing the GRS is worrying. If I understand correctly, the effect sizes used for weighting were obtained from analyses of the ALSPAC dataset, i.e. the same dataset that was utilized in the present analyses. This leads to a bias where the effect of the GRS on BMI is inflated. To construct the GRS correctly, the weights would need to be extracted from the original GWAS discovery study or another large, independent dataset. Alternatively, an unweighted GRS could be used.

2. The authors have combined results from 97 independent variants only, rather than taking advantage of more recent latest GWAS including >400 independent variants [PMID 30239722].

3. The authors state hat “the effect of the GRS is not well studied in children” (p. 3) and that “the role of the obesity-specific GRS remains largely unknown in children and adolescents” (p. 15). However, many studies of obesity-specific GRS in children have been published [e.g. PMID 29211904, 30515969, 28008729, 24244521]. Thus, these claims do not seem valid.

4. The authors report having evaluated the interaction between the duration of exclusive breastfeeding and GRS on BMI growth trajectories. However, they have not performed a formal test for interaction, but rather base their conclusions on observing values between stratified subgroups, which does not seem sufficient statistical evidence for an interaction. The authors should perform a formal test for interaction by including an interaction term in the model. Similar issue applies to e.g. p. 14 where the authors state that “a duration of EBF for 3 months had significantly less impact in decreasing BMI than 5 months of EBF”. A formal test for the difference between the groups should be included here, to make such claim about statistical significance.

Reviewer #3: In this paper, the authors examine whether exclusive or any breastfeeding up until 5 months of age is protective against later BMI gain by BMI-associated genetic risk. The paper is in general well-written and the analyses are performed satisfactorily. Additionally, the findings are of broad public interest, and the conclusions are appropriate.

I do have some minor comments:

1. It is fine to use only 97 BMI-associated variants, but then this choice should be justified, given that over 900 are now published that explain ~6% of the variance (Yengo HMG 2018). This should especially be considered in the Discussion, and perhaps expanding the GRS could be a "future direction".

2. The abstract could use some clarification. Obesity is now better understood compared to what? I also find the second sentence unclear. Please also briefly add some detail about the GRS used in this study in the abstract, for example, how many SNPs, associated with adult or childhood BMI originally, etc.

3. Is there a word "and" missing in the final sentence of the abstract, "EBF influences early life growth AND development"?

4. It seems an overstatement to say that EBF plays a "critical role" based on this study's findings (only a 1.4kg/m2 decrease in BMI in boys at 18), but certainly EBF is important and has lifelong benefits on BMI gain, especially for those with high genetic load of increasing alleles.

5. In the author summary, saying something is "perfect" is very subjective. Please rephrase using objective language.

6. Author summary, last sentence: please change "susceptibility alleles" to "increasing alleles"

7. Introduction, line 75: It is confusing and misleading to say that BF extends "beyond healthy children, e.g., in children with higher genetic risks". Kids with GRS for high BMI can still be healthy.

8. Throughout the paper, please say "pediatric BMI" instead of "child BMI" since you are also talking about adolescents.

9. Page 5, line 87: please add whether the GRS was weighted or unweighted

10. Line 90: At what age did the GRS explain 1.5% of BMI variability?

11. Page 8, line 146: Change "principle" to "principal"

12. I question whether it makes sense to describe the equivalence of a 1-unit increase in GRS in terms of NUMBER of effect alleles carried given that the GRS was weighted..

13. Page 9, line 174: "maternal preconception" is perhaps missing a word?

14. Line 173: categorize  categorizeD

15. I think Table 2 and 3 would be easier to grasp with graphical representation, and put the data in these tables in the supplement. Why not give exact p-values in the table? Also please check the formatting-- there should be a space after a numeral and before a parenthesis.

16. Discussion, line 284: "shed lights"  shed light

17. Sentence on line 284 is confusing. I would remove "The value of" and "in life" -- it is clearer without those

18. Line 291: it  they

19. Top of page 16, please discuss why this GRS was used and not most recent data from Yengo et al included.

20. Line 301: "essential fluid" sounds odd

21. Finally, the English is overall good but requires careful editing by a native speaker. There are many small mistakes throughout.

**Have all data underlying the figures and results presented in the manuscript been provided?**

Reviewer #1: Yes

Reviewer #2: Yes

Reviewer #3: Yes

PLOS authors have the option to publish the peer review history of their article (what does this mean?). If published, this will include your full peer review and any attached files.

Reviewer #1: No

Reviewer #2: No

Reviewer #3: No

---

## [Decision Letter · Decision Letter 1]

22 Apr 2020

Dear Dr Briollais,

We are pleased to inform you that your manuscript entitled "Exclusive breastfeeding can attenuate body-mass-index increase among genetically susceptible children: a longitudinal study from the ALSPAC cohort" has been editorially accepted for publication in PLOS Genetics. Congratulations!

Yours sincerely,

Samuli Ripatti

Associate Editor

PLOS Genetics

Gregory Barsh

Editor-in-Chief

PLOS Genetics

Comments from the reviewers (if applicable):

Reviewer's Responses to Questions

**Comments to the Authors:**

Reviewer #1: The results presented in the manuscript are now on much more solid basis as the GRS weights are now derived from external data set. Luckily, this didn't change the results in a big picture. I have no more comments and I think that the manuscript should be considered for publication in Plos Genetics.

Reviewer #2: The authors have addressed my comments appropriately. I have no further remarks.

Reviewer #3: Thank you, all of my concerns have been addressed.

**Have all data underlying the figures and results presented in the manuscript been provided?**

Reviewer #1: Yes

Reviewer #2: Yes

Reviewer #3: Yes

PLOS authors have the option to publish the peer review history of their article (what does this mean?). If published, this will include your full peer review and any attached files.

Reviewer #1: No

Reviewer #2: No

Reviewer #3: Yes: Diana L. Cousminer

**Data Deposition**

http://datadryad.org/submit?journalID=pgenetics&manu=PGENETICS-D-20-00028R1

**Press Queries**

---

## [Editor Report · Acceptance letter]

19 May 2020

PGENETICS-D-20-00028R1 

Exclusive breastfeeding can attenuate body-mass-index increase among genetically susceptible children: a longitudinal study from the ALSPAC cohort 

Dear Dr Briollais, 

We are pleased to inform you that your manuscript entitled "Exclusive breastfeeding can attenuate body-mass-index increase among genetically susceptible children: a longitudinal study from the ALSPAC cohort" has been formally accepted for publication in PLOS Genetics! Your manuscript is now with our production department and you will be notified of the publication date in due course.

With kind regards,

Matt Lyles

PLOS Genetics

On behalf of:
